# Targeting Transferrin Receptor 1 for Enhancing Drug Delivery Through the Blood–Brain Barrier for Alzheimer’s Disease

**DOI:** 10.3390/ijms26199793

**Published:** 2025-10-08

**Authors:** Xinai Shen, Huan Li, Beiyu Zhang, Yunan Li, Zheying Zhu

**Affiliations:** School of Pharmacy, The University of Nottingham, Nottingham NG7 2RD, UK; xinai.shen@nottingham.ac.uk (X.S.); paxbz1@exmail.nottingham.ac.uk (B.Z.);

**Keywords:** transferrin receptor 1, blood–brain barrier 2, Alzheimer’s disease 3

## Abstract

Drug delivery to the brain faces a critical obstacle in the form of the blood–brain barrier (BBB), which severely limits therapeutic options for Alzheimer’s disease (AD). Transferrin receptor 1 (TfR1) is abundantly expressed in brain capillary endothelial cells, offering a potential pathway for circumventing this barrier. Physiologically, TfR1 binds to iron-laden transferrin, leading to cellular uptake through clathrin-mediated endocytosis. Within acidic endosomes, the iron is released, and the receptor–apotransferrin complex recycles to the cell surface for further rounds of transport. Furthermore, studies in AD mouse models have demonstrated that TfR1 expression in brain microvessels remains stable, highlighting its suitability as a delivery target even in disease conditions. Based on this, various drug delivery strategies targeting TfR1 have been developed, including bispecific antibodies, antibody fragments, ligand conjugates, and nanoparticle-based carriers. While these approaches hold great promise, they face practical limitations such as competition with endogenous transferrin, receptor saturation, and inefficient intracellular trafficking. This review details the current understanding of TfR1-mediated BBB transport mechanisms, evaluates emerging delivery platforms, and argues that TfR1 represents an accessible gateway for brain-targeted therapeutics in AD. The insights presented will be of interest to researchers in molecular biology, pharmacology, and drug development.

## 1. Introduction

Alzheimer’s disease (AD) is the most prevalent neurodegenerative disorder globally, affecting millions of people and placing an immense burden on healthcare systems and societies [1]. The hallmark of AD is the accumulation of amyloid-beta (Aβ) plaques and neurofibrillary tangles composed of hyperphosphorylated tau protein in the brain, which may clinically manifest as progressive memory loss, cognitive decline, and behavioural changes [2]. Despite ongoing research into the pathogenesis of AD, including the amyloid cascade hypothesis and the role of neuroinflammation, significant unmet medical needs remain [3]. One of the major challenges in AD therapy development is the difficulty of drug delivery to the brain. Due to the unique physiological characteristics of the central nervous system (CNS), especially the existence of the blood–brain barrier (BBB), many medicines are difficult to deliver into the brain [4,5].

The BBB is a highly specialised and dynamic interface composed of brain capillary endothelial cells (BCECs) interconnected by tight junctions (TJs), along with pericytes, astrocytes, and neurons, collectively forming the neurovascular unit (NVU) [6]. The strict selectivity of the BBB restricts most small-molecule drugs and almost all large-molecule pharmaceuticals from entering the CNS and reaching therapeutic concentrations [7]. This inherent tight junction is a primary reason for the high failure rate of CNS drug development, resulting in many AD candidate therapeutic drugs failing to reach their therapeutically relevant concentrations, targeting the related lesion sites in the brain [4].

In recent years, a great deal of research effort has been devoted to innovative strategies to bypass or cross the BBB efficiently and safely. Among these strategies, harnessing endogenous receptor-mediated transcytosis (RMT) pathways has emerged as a promising non-invasive approach [8]. RMT involves the specific binding of ligands to receptors expressed on the surface of BCECs, followed by their internalisation and intracellular trafficking, and the ligand–receptor complexes are then released into the brain parenchyma [9]. These receptors are naturally employed to transport essential nutrients and macromolecules into the brain [10].

The transferrin receptor 1 (TfR1) has attracted much attention due to its high expression on the surface of BCECs and its crucial role in iron homeostasis by mediating the uptake of iron-bound transferrin into cells [11]. This review aims to provide a comprehensive overview of the current understanding of targeting TfR1 to enhance drug delivery across the BBB for the treatment of AD. We will explore the structure and function of the BBB under physiological conditions and AD, summarise the biology of TfR1, and evaluate multiple strategies for brain drug delivery via TfR1.

## 2. The Blood–Brain Barrier in Health and Alzheimer’s Disease

### 2.1. Blood–Brain Barrier Structure and Function

The BBB is a highly specialised and dynamic neurovascular interface that plays a critical role in maintaining the homeostatic environment of the CNS [6]. It is formed primarily by BCECs that are tightly conjugated and ensheathed by pericytes, astrocytes’ end-feet, and a basement membrane, these together constituting the NVU. This structure allows for extremely low paracellular permeability inside and outside the brain, while precisely regulating intercellular exchange to maintain neural homeostasis. As in other epithelial cells, the TJ backbone in brain endothelial cells is composed of transmembrane proteins (occludin, claudins, and junctional adhesion molecules (JAMs)), which recruit numerous membrane-associated cytoplasmic proteins [12]. These proteins limit paracellular flux, whereas selective transcellular routes support nutrient delivery and waste removal. The routes include solute carrier–mediated transport (e.g., GLUT1 for D-glucose and LAT1 for large neutral amino acids), ATP-binding cassette efflux pumps (ABCB1/P-glycoprotein, ABCG2/BCRP, MRPs) for excreting foreign compounds and metabolites, and vesicular pathways [13]. Astrocytic end-feet polarise aquaporin-4 (AQP4) water channels and, in conjunction with the perivascular spaces, support potassium spatial buffering and calcium signalling, complementing BBB efflux [14]. From a physiological perspective, this design ensures a stable ionic environment and nutrient supply while preventing the entry of pathogens and toxins. Barrier selectivity is not static; it is dynamically regulated by the NVU to maintain perfusion, glucose or oxygen delivery, and metabolite clearance despite changes in neuronal load [15]. BBB integrity and neuronal viability are crucial in humans.

### 2.2. Blood–Brain Barrier Dysfunction in AD

BBB disruption in AD is characterised by several pathological hallmarks, including the breakdown of TJs, increased permeability, reduced cerebral blood flow, and impaired transport of Aβ and other metabolites. Increasing evidence suggests that BBB dysfunction can be detected in the preclinical and early symptomatic stages of AD [16]. Dynamic contrast-enhanced magnetic resonance imaging (DCE-MRI) reveals that, in the early stages of AD or mild cognitive impairment, the permeability of the BBB in the cerebral cortex and deep grey matter (as indicated by the Ki value) is increased, and the degree of permeability is positively correlated with cognitive decline [17]. Even in individuals with normal levels of Aβ and tau proteins, BBB disruption in the hippocampus can be observed in those with early cognitive impairment. These abnormalities primarily occur in the hippocampus and adjacent medial temporal lobe regions, while the extent of disruption in the cortex and deep grey matter varies regionally. Even in cognitively normal older adults, regional differences in permeability maps suggest heterogeneity in BBB integrity, which changes before the onset of clinical symptoms [18]. Fluid biomarkers further confirm this early signal and help pinpoint the cellular location of the damage [19]. For example, the CSF biomarker for pericyte damage, soluble PDGFRβ (sPDGFRβ), is elevated in older adults and MCI patients, positively correlating with DCE-MRI leakage and classic BBB permeability markers (such as the CSF and serum albumin ratio Qalb and CSF fibrinogen), even when amyloid or tau levels are normal, suggesting that cognitive decline can lead to its elevation [20,21]. Other vascular markers (e.g., ANGPT-2) are also associated with BBB permeability and tau-related damage [22].

Studies have revealed multiple mechanisms that contribute to BBB dysfunction. Pericyte degeneration is particularly pronounced in individuals with AD and those with the *APOE4* gene. APOE4 weakly binds to endothelial cell lipoprotein receptor-related protein 1 (LRP-1), leading to activation of the nuclear factor-κB (NF-κB)-matrix metalloproteinase-9 (MMP-9) pathway in pericytes, accelerating the degradation of tight junction proteins and basement membrane proteins, leading to increased permeability [23,24,25]. This pathway links genotype to early vascular damage, independent of plaque or tangle burden. APOE also further influences tau pathology and neuroinflammation [26]. Pericyte degeneration impairs the control of cerebral blood flow and neurovascular coupling by capillaries, reduces oxygen supply, and exacerbates neuronal metabolic stress. These effects can occur before overt neurodegeneration and further amplify the downstream amyloid and tau cascade [27]. In in vivo models, APOE exacerbates tau-induced neurodegeneration, independently of Aβ, and promotes a more neurotoxic microglia phenotype [28]. Furthermore, APOE4 alters mitochondrial metabolism, increases oxidative stress, and disrupts lipid raft composition, further impairing receptor signalling at synapses [29]. TJ proteins are selectively reduced in the cortex of AD [30]. Postmortem analysis of multiple brain regions revealed decreased levels of claudin-5 and occludin in AD patients, which correlated with synaptic markers and, in some individuals, was independent of insoluble plaque or tangles [31]. In addition, LRP-1 mediates the efflux of Aβ from the brain, while the receptor for advanced glycation end products (RAGE) promotes the influx of Aβ. Imbalance in the function of these two factors can disrupt the junctional complex and barrier function, leading to increased Aβ accumulation in the brain [32]. Oxidative stress can also induce changes at the molecular level. Reduced blood flow and impaired capillary perfusion lead to insufficient oxygen supply to endothelial cells, resulting in hypoxia, which stabilises HIF-1α and increases reactive oxygen species (ROS) levels [33]. These redox signals can slow down clathrin-mediated endocytosis and receptor recycling and exacerbate AQP4 mislocalisation and extracellular matrix remodelling [34]. In summary, early pathological changes in the BBB can be detected, and these changes are not uniformly distributed across different regions; the underlying mechanism primarily involves signalling pathways between pericytes and endothelial cells. Although BBB dysfunction has been observed in AD patients, it is crucial to note that this disruption is usually localised and heterogeneous, while for most large molecules, particularly biopharmaceuticals, the barrier remains largely intact, thus requiring targeted delivery strategies [35]. This further supports the rationale for using targeted drug delivery strategies, rather than simply assuming that there is a general increase in BBB permeability in early AD.

Consistent with the DCE-MRI and fluid-biomarker signal of early, regionally heterogeneous BBB compromise, BBB dysfunction involves all major components of the neurovascular unit. Selective changes in endothelial cells affect energy supply and amyloid transport. Specifically, reduced expression of GLUT1 in the blood vessels of AD patients is associated with cognitive decline [36]. Verapamil PET imaging reveals decreased P-glycoprotein function in endothelial cells, consistent with impaired endothelial clearance function [37]. Endothelial LRP1 expression decreases with age and disease progression, while RAGE promotes inflammatory signalling and amyloid influx [38].

TJ architecture provides the structural substrate for the subtle permeability increases seen on imaging, with region-specific loss or mislocalisation of claudin-5 and occludin in cortex. Pericytes form an early, sensitised locus of injury, reflected by higher CSF soluble PDGFRβ that tracks DCE-MRI leakage and barrier indices and aligns with reduced coverage and junctional degradation [39]. Astrocytic end-feet exhibit AQP4 depolarisation away from the perivascular membrane, slowing glymphatic and perivascular clearance and associating with higher amyloid burden [40]. Oxidative stress subject to astrocytes causes DNA damage, resulting functional impairment, which reduces their support for the BBB [41]. Basement membranes thicken and shift in composition with ageing and AD and in cerebral amyloid angiopathy, impeding intramural periarterial drainage and favouring vascular amyloid deposition [42]. These observations together explain why BBB disruption in AD is characterised by an early stage but regional distribution.

## 3. Transferrin Receptor 1: Biology and Function

### 3.1. Transferrin Receptor 1 Structure and Function

TfR1, also known as CD71, is a homodimeric type II transmembrane glycoprotein that plays a central and indispensable role in cellular iron uptake and metabolism by binding diferric transferrin (holo-Tf) at the cell surface and internalising the complex through clathrin-mediated endocytosis [43]. TfR1 is the primary gateway for cells to acquire iron from the bloodstream, where iron is transported in its ferric (Fe^3+^) state, tightly bound to the glycoprotein transferrin (Tf).

Each TfR1 monomer is composed of a short N-terminal intracellular region, a transmembrane domain, and a large extracellular ectodomain to form the canonical transferrin-binding site [44]. The cytosolic tail contains a conserved tyrosine-based internalisation motif (YTRF) that recruits the AP-2/clathrin machinery and drives rapid endocytosis. Mutational analyses indicate that the YTRF sequence is both necessary and sufficient for efficient uptake [45]. The ectodomain, which is responsible for binding transferrin, consists of three distinct domains: the protease-like domain, the apical domain, and the helical domain [46]. The binding of two iron-containing transferrin molecules to the TfR1 homodimer initiates a highly efficient process known as receptor-mediated endocytosis. Upon binding, the Tf-TfR1 complex is internalised into clathrin-coated vesicles, which then mature into early endosomes. Within the acidic environment of the endosome (pH ~5.5), a conformational change occurs in both transferrin and TfR1, leading to the release of iron from transferrin [47]. As illustrated in Figure 1, the released ferric iron is then reduced to ferrous iron (Fe^2+^) by STEAP ferrireductases and transported into the cytoplasm via divalent metal transporter 1 (DMT1) [48]. Crucially, apotransferrin (iron-free transferrin) remains bound to TfR1 at low endosomal pH after iron release. This apotransferrin-TfR1 complex is then recycled back to the cell surface, where, upon exposure to neutral extracellular pH, apotransferrin dissociates from the receptor, and TfR1 becomes available for another round of iron uptake [49]. In brain endothelial cells, a portion of cytosolic Fe^2+^ is exported across the abluminal membrane by ferroportin and re-oxidised by GPI-anchored ceruloplasmin to Fe^3+^ to reload interstitial transferrin, while TfR1-bound ligands can also undergo vectorial transcytosis to the abluminal side (Figure 1). This recycling mechanism is highly efficient, ensuring continuous iron supply without significant degradation of the receptor.

The expression of TfR1 is tightly regulated and varies across different cell types and physiological conditions. It is highly expressed on the luminal surface of BCECs, which act as gatekeepers of the BBB [50]. This preferential expression of TfR1 on brain endothelial cells compared to peripheral endothelial cells makes it an attractive and relatively specific target for delivering substances across the BBB into the CNS [51]. The physiological role of TfR1 in mediating essential iron transport across the BBB underscores its robust and active transcytosis pathway, which can be exploited to deliver drugs for the treatment of AD.

### 3.2. Transferrin Receptor 1 in Alzheimer’s Disease

The complex relationship between iron homeostasis and neurodegenerative diseases, particularly in AD, has attracted increasing attention, making the status of TfR1 in AD pathology a key area of investigation for drug delivery strategies. Given the central role of TfR1 in iron uptake and its high expression at the BBB, understanding whether its function or expression is altered in AD is crucial for evaluating its viability as a therapeutic or delivery target. 

One of the most significant findings regarding TfR1 in AD is that its expression levels and functional integrity at the BBB appear to be largely preserved despite the presence of AD neuropathology. This is a crucial distinction, as many other BBB components, such as tight junctions and efflux transporters, exhibit dysfunction or altered expression in AD, potentially complicating drug delivery efforts that rely on their normal function. A comprehensive study by Bourassa et al. directly addressed this concern by investigating TfR1 levels and TfR-mediated uptake in both human post-mortem brain tissue and transgenic mouse models of AD [52]. Their findings demonstrated no significant difference in TfR1 protein levels in whole homogenates from human post-mortem parietal cortex and hippocampus between individuals with and without a neuropathological diagnosis of AD. Similarly, TfR1 concentrations in human brain microvessels isolated from the parietal cortex were comparable between control and AD cases [53]. This consistency was further corroborated in murine models, where TfR1 levels in isolated brain microvessels were not significantly different between 12- and 18-month-old NonTg and 3xTg-AD mice [52]. This is a crucial distinction, as many other BBB components, such as TJs and efflux transporters, exhibit dysfunction or altered expression in AD, which can complicate the delivery of drugs that rely on their normal function [54]. Because of this property, many large molecules have been attempted to be transported from the blood to the brain via the TfR1 pathway [55]. 

Beyond mere expression levels, the functional integrity of TfR1-mediated transport is equally critical. Bourassa et al. conducted in situ brain perfusion assays to measure the brain uptake and internalisation of a fluorolabelled TfR1-targeting monoclonal antibody (Ri7) into BCECs [52]. Consistently, TfR1-mediated uptake in BCECs was found to be similar between 3xTg-AD mice and non-transgenic controls across 12, 18, and 22 months. Fluorescence microscopy analysis following intravenous administration of the fluorolabeled Ri7 antibody further highlighted its widespread distribution throughout the cerebral vasculature, with no significant signal observed in neurons or astrocytes, indicating its primary association with the endothelial cells of the BBB. Human post-mortem microvessels and transgenic mouse models show preserved TfR1 abundance and TfR1-dependent internalisation into brain endothelium relative to controls, with increased cortical levels in some models, indicating that the receptor pathway itself is largely intact as other endothelial systems falter [52]. These collective data strongly suggest that both TfR1 protein levels and TfR1-dependent internalisation mechanisms are preserved in the presence of Aβ and tau neuropathologies, providing strong evidence supporting the potential of TfR1 as a vector target for drug delivery into BCECs in AD. In early symptomatic AD cohorts, the TfR1-shuttled anti-Aβ bispecific trontinemab (RG6102) achieves rapid, deep amyloid-PET clearance at low systemic doses with 91% of participants PET-negative at 28 weeks on 3.6 mg/kg, 72% reaching ≤11 centiloids and low rates of ARIA-E, consistent with functional receptor-mediated transcytosis in patients despite oxidative and vascular stressors [56]. The delivery implications are practical and can be validated. The priority of monovalent, moderate-affinity, and pH-tuned TfR1 engagement can minimise receptor residence time and competition with transferrin, incorporate perfusion limitations and capillary transit-time heterogeneity into exposure modelling, and account for slower parenchymal clearance where AQP4 polarity and basement-membrane mechanics are altered.

Against this background, the cellular pathway of TfR1 at the luminal surface of brain endothelium provides a mechanistic rationale for its use. TfR1 supports a continuous cycle of clathrin uptake, early endosomal sorting, and rapid recycling to the surface [57]. The endocytosed Tf-TfR1 complex moves to Rab5 early endosomes, transferrin releases iron upon lumen acidification, and the receptor returns to the plasma membrane via recycling endosomes without significant degradation [58,59,60]. Given this background, the cellular itinerary of TfR1 at the luminal surface of brain endothelium provides a mechanistic rationale for its use. Because this pathway has been mapped in detail in multiple cell systems, it enables predictable routes for drug delivery design and contrasts with the less well-defined or less reliable routes used by many alternative BBB targets [61]. 

The TfR1 expression pattern in AD amplifies this advantage. In some models, endothelial and cortical TfR1 levels are even elevated [53,62]. In contrast, other canonical BBB receptors exhibit disease-associated instability. Endothelial LRP1, a major efflux pathway for Aβ, declines with age and is further reduced in human vasculature and mouse models of AD, impairing Aβ clearance from the brain to the blood [63,64,65]. GLUT1, the major glucose transporter in brain microvasculature, is consistently reduced in AD patient microvasculature and experimental systems, and these reductions are associated with vascular dysfunction and cognitive decline [66,67,68]. AD also exhibits defects in cerebrovascular insulin receptors and lower levels of INSRα-B, suggesting impaired and potentially altered IR-based shuttling [69]. These observations suggest that the location of TfR1 at the apical membrane is maintained during disease progression, whereas LRP1, GLUT1, and INSR may decline, thereby reducing the viability of ligands that depend on these targets. Furthermore, TfR1 is highly expressed in the brain capillary, which increases drug exposure throughout the brain.

However, the dysregulation of iron homeostasis in AD brings uncertainty to the targeted delivery of TfR1. While TfR1 expression at the BBB appears stable, other components of the iron regulatory machinery may be altered. A study has investigated the levels of calcium/calmodulin-dependent protein kinase kinase 2 (CAMKK2), Tf, and TfR1 proteins in the temporal cortex of post-mortem AD patients, which revealed a significant decrease in CAMKK2, Tf, and TfR1 levels in AD patients’ temporal cortices compared to cognitively normal individuals, independent of age or post-mortem interval [70]. Furthermore, increased iron content in AD brains was significantly correlated with reduced Tf/TfR1 protein levels. This suggests that while TfR1 at the BBB might be preserved, its overall levels in specific brain regions, potentially within neurons or glia, could be downregulated, contributing to iron overloading and neurodegeneration through iron-induced toxicity. The proposed mechanism involves CAMKK2 downregulation disrupting Tf/TfR1 signalling, leading to the enhanced clearance or post-trafficking degradation of Tf/TfR1. This finding highlights a potential complexity: while TfR1 on the endothelial cells of the BBB remains a viable target for transcytosis, the downstream effects on parenchymal cell iron homeostasis and TfR1 expression within the brain itself warrant further investigation for comprehensive AD therapy. Nevertheless, for the specific purpose of crossing the BBB, the evidence supports TfR1’s reliability as a target in AD.

## 4. Transferrin Receptor 1-Mediated Drug Delivery Strategies and Challenges

The BBB is inherently impermeable to most therapeutic drugs, necessitating innovative strategies for the effective delivery of drugs to the CNS, particularly for complex neurodegenerative diseases such as AD. Among the various approaches, exploiting the endogenous RMT pathway via TfR1 has emerged as a promising approach. This section will explore various TfR1-mediated drug delivery strategies, including antibody-based strategies and ligand-based strategies (Figure 2), and discusses the challenges associated with each approach.

### 4.1. Receptor-Mediated Transcytosis Principles

RMT is a highly regulated cellular process initiated by ligand attachment to specific receptors on the luminal side of vascular endothelial cells, triggering membrane invagination and subsequent endocytosis. In brain endothelial cells, RMT binding occurs on the luminal membrane, and the ligand usually binds to receptors enriched on brain endothelial cells [71]. The ligand–receptor complexes are then internalised by clathrin into Rab5-positive early endosomes, where V-ATPase-driven acidification changes the strength of the interaction between the receptor and bound ligand [72]. Complexes that dissociate at this stage are captured by narrow sorting tubes that bud from early endosomes, penetrate the thin endothelial cytoplasm, and fuse with the luminal membrane, releasing the cargo into the perivascular space, while the receptors recycle to the luminal surface [73]. In the healthy brain, caveolar traffic is actively inhibited by Major facilitator superfamily domain-containing 2a (MFSD2A), such that clathrin entry and endosomal tubulation dominate efficient RMT [74]. These steps have been established that endosomal pH, tubule formation, and receptor recycling collectively determine the efficiency of transcytosis across the BBB.

Recent advancements in in vitro human BBB models offer a platform for evaluating targeting of the TfR1 under continuous perfusion and physiological shear stress. Human induced pluripotent stem cell-derived models and microvessel systems recapitulate TJs, efflux activity, and luminal-to-abluminal polarity, while maintaining optical access for live imaging. Protocols for self-assembled microvascular networks describe device culture timelines, permeability measurements within the physiological range, and standardised imaging procedures, enabling quantitative assessment of receptor-mediated transport under conditions closer to the human BBB. Open microfluidic microvessel designs further enhance these capabilities by allowing direct luminal access, unidirectional flow, and high-resolution confocal imaging of endocytosis and vesicular trafficking. A representative human brain endothelial microvessel-on-a-chip model exhibited stable barrier characteristics under flow and supported real-time visualisation of intracellular trafficking, valuable for distinguishing clathrin-mediated endocytosis, early endosome acidification, and recycling pathways during transferrin receptor binding [75].

Platforms optimised for larger biotherapeutics have begun to report antibody movement across human endothelium together with barrier integrity readouts. In a perfused human barrier on a chip, penetration of an antibody that binds the human transferrin receptor exceeded that of a control antibody, with apparent permeability values of approximately 2.9 × 10^−5^ vs. 1.6 × 10^−5^ cm per minute, which demonstrates sensitivity to saturable transport while maintaining low passive leak. This format also supports multi-chip layouts that increase throughput for ligand ranking [76].

Experiments that pair engineered ligands with human transferrin receptor further illustrate the utility of these systems. A computationally designed small protein with nanomolar affinity for the human receptor crossed a human organ on a chip barrier, confirming that epitope selection and affinity can be optimised against a human endothelium context before animal studies [77]. Complementary work on iPSC-based barrier chips reported physiologically relevant electrical resistance and accurate prediction of drug permeability, and showed that whole blood perfusion can be tolerated while neural elements remain protected, which broadens the set of safety and compatibility readouts [78].

These devices also inform transport kinetics and saturation. Under flow, endothelial alignment reduces nonspecific uptake and improves the dynamic range for concentration–response experiments, which supports the estimation of apparent capacity and affinity parameters for transferrin receptor-mediated transcytosis. Protocols describe permeability assays that can be completed within practical timelines and can be combined with high content imaging to track endosomal routing, recycling and lysosomal engagement as dose increases, thereby identifying the exposure zones that precede receptor saturation [79]. 

Safety can be examined in parallel with efficacy in these platforms. Perfused chips permit simultaneous monitoring of TJ continuity, recovery of electrical resistance after exposure and inflammatory responses that accompany barrier stress. Studies report decreases in electrical resistance and increases in tracer permeability during inflammatory challenge with associated immune cell adhesion and extravasation, which establishes benchmarks for acceptable perturbation during testing of transferrin receptor shuttles [80]. These coupled readouts help select designs with moderate apparent affinity, efficient endosomal dissociation and minimal barrier disruption prior to first in human dose setting. 

Hence human barrier on a chip and microvessel models now provide a physiologically informed setting in which transferrin receptor targeting can be evaluated under shear with direct optical access. These systems support the head-to-head comparison of ligand density, affinity and valency across identical flow and matrix conditions, allow direct measurement of antibody or nanoparticle passage, and provide concurrent safety metrics. The emerging literature therefore strengthens the translational chain by linking molecular design to transport efficiency, receptor saturation behaviour and endothelial tolerance in a human context [79].

For TfR1, the process begins with the binding of iron-loaded transferrin (or a therapeutic agent engineered to mimic transferrin) to TfR1 on the luminal surface of the BCECs. This binding triggers the formation of clathrin-coated pits, leading to the internalisation of the receptor–ligand complex into endosomes. In early endosomes, the drop in pH weakens interactions for antibodies designed with pH-dependent binding, so the ligand-attached molecule disengages from TfR1 and can enter sorting tubules [81]. Monovalent engagement and moderate affinity reduce receptor clustering, which increases the fraction routed into tubules and favours abluminal exocytosis and release into the brain interstitium [82]. When the affinity is too high or the binding is bivalent cross-linked, the cargo accumulates in the endolysosomal compartment, and transcytosis decreases. When the affinity, valency, and pH are all within the appropriate range, the delivered complex is translocated from the lysosome to the abluminal tubule [71]. These elements provide a reference direction for the design of ligands delivered via the RMT pathway.

An expanded view places endothelial transcytosis within a fluid dynamic landscape in which perivascular and glymphatic pathways shape residence time and spatial distribution after abluminal release [83]. Convective exchange along periarterial and perivenous routes depends on aquaporin 4 at astrocytic end-feet [84]. Loss of perivascular aquaporin 4 polarisation slows solute clearance, reduces beta amyloid efflux and accelerates plaque formation in rodents, which links fluid movement after BBB entry to AD relevant pathology [85].

Complementary evidence comes from imaging tools that quantify glymphatic function in living systems [86]. In humans, diffusion tensor image analysis along perivascular spaces provides a noninvasive index and shows impairment in mild cognitive impairment and AD cohorts. Intrathecal contrast enhanced magnetic resonance imaging demonstrates tracer access from cerebrospinal fluid into brain tissue and subsequent drainage toward cervical lymph nodes with peak enhancement near 24 h, findings that align with perivascular clearance routes [87].

These observations indicate that schedules, exposure models and imaging readouts for TfR1 targeted agents are better interpreted within a framework that couples endosomal itinerary and receptor saturation with downstream clearance [88]. Sleep state and neuromodulatory tone modulate bulk fluid movement and waste removal, and experimental as well as theoretical work links vascular pulsation and time of day to changes in perivascular exchange, which can shift effective residence time after entry [89]. Integrating these measurements with receptor-mediated transcytosis parameters situates TfR1 delivery within brain fluid dynamics and provides a basis for stage appropriate dose selection and interpretation of regional distribution.

### 4.2. Antibody-Based Strategies

Antibody-based strategies leverage the high specificity and affinity of monoclonal antibodies (mAbs) to target TfR1 and facilitate BBB crossing [90]. This approach often involves engineering therapeutic antibodies such as bispecific antibodies or fusion proteins, where one arm targets TfR1 and the other targets a therapeutic molecule or an AD-related pathology [91]. A common application is for BBB-targeted bispecific antibody carrying a therapeutic cargo to bind to receptor-mediated transcytosis (RMT, e.g., TfR1) receptors on the luminal surface of brain endothelial cells. The antibody–receptor complex is then endocytosed and transported to the abluminal side, thereby delivering the cargo into the brain parenchyma [59]. Other vector–cargo formats can exploit the same transcytosis mechanism.

To direct address comparative performance, antibody shuttles generally achieve higher absolute brain exposure than ligand-based nanoparticles in matched species and dosing paradigms while also offering a longer systemic persistence due to the Fc region [82]. In non-human primate models and modelling analyses, TfR1-mediated antibody shuttling significantly increased brain parenchyma exposure without increasing cerebrospinal fluid distribution, suggesting a lower dose range required for plaque clearance in humans. Early clinical data from the Brainshuttle AD study (NCT04639050) showed that a 3.6 mg/kg dose led to amyloid negativity in most subjects within 28 weeks, indicating a clear pharmacodynamic advantage of this platform in humans [92]. 

#### 4.2.1. Monoclonal Antibodies

Anti-TfR antibodies have been used as trafficking modules to pull large amounts of cargo across the BBB via selective receptor-mediated transcytosis. On the luminal surface, anti-TfR IgG binds to TfR1 and triggers clathrin uptake. Because high binding affinity is undesirable, optimising binding affinity is a key consideration during design. In the mouse, monovalent low or mid affinity engagement of TfR improved target engagement in brain by about 55-fold compared with a high avidity format. This result aligns with the observed increase in brain exposure when the parental IgG is converted to a shuttle [82]. At the same time, peripheral TfR1 on reticulocytes and other cells can drive off target binding and target mediated clearance. In a human BBB transcytosis assay, antibodies that lose affinity at pH 5.5 exhibited increased transcytosis and reduced endosomal retention compared to pH-independent binders with similar neutral pH affinity [93]. Practical comparisons show that formats enforcing one-to-one engagement per TfR1 dimer perform well. For example, scFv8D3 units separated by short linkers behave functionally as monovalent binders and yield higher brain uptake. Allowing the same binder to engage both protomers on a dimer creates an avid bivalent complex that tends to be retained and routed to degradative compartments, which lowers delivery [94]. Taken together, monovalent, moderate-affinity and pH-attenuating formats are preferred for transcytosis, whereas bivalency is generally disfavoured unless affinity is deliberately reduced to offset avidity [94].

Practical comparisons show that formats enforcing one-to-one engagement per TfR1 dimer. For example, scFv8D3 units separated by short linkers behave functionally as monovalent binders and yield higher brain uptake. Allowing the same binder to engage both protomers on a dimer creates an avid bivalent complex that tends to be retained and routed to degradative compartments, which lowers delivery [94]. Taken together, monovalent, moderate-affinity and pH-attenuating formats are preferred for transcytosis, whereas bivalency is generally disfavoured unless affinity is deliberately reduced to offset avidity.

mAbs offer precise epitope selection, scalable production, and a long serum half-life. However, peripheral TfR1 on reticulocytes and other cells can impair antibody binding efficiency to the desired target, particularly for high-affinity, effector-competent IgGs [95]. To minimise this, a series of design strategies is employed, including adjusting affinity to the mid-range, enforcing monovalent binding, engineering pH-dependent attenuation within endosomes, and inhibiting Fc effector function [96,97,98]. These strategies reduce reticulocyte effects while preserving microglial engagement of the therapeutic arm to its brain target.

#### 4.2.2. Antibody Fragments

Fragments shrink the transport module and generally improve parenchymal dispersion after transcytosis. ScFv shuttles derived from the murine 8D3 TfR1 antibody have been repeatedly fused to CNS-targeted binders [99]. In mice, anti-Aβ affibodies fused to the scFv8D3 efficiently reach the parenchyma but are rapidly washed out when the affinity balance between Aβ and TfR1 is suboptimal [100]. Therefore, antibody design and selection must balance transport efficiency with clearance time. Related 8D3-based scFv fusions have delivered enzymes or neutralising payloads to the brain, demonstrating that compact TfR-binding fragments can drive efficient transcytosis if valency and pH behaviour are tuned [101].

Nanobodies and single variable domain on heavy chains (VNARs) are even smaller and can be selected for cross-species TfR1 reactivity [102]. The shark VNAR TXB2 rapidly crosses the BBB, is brain-selective, and can carry a variety of cargoes. Variants of TXB2 further improve transport while avoiding competition with transferrin [95]. Mouse TfR-binding nanobodies engineered with histidine attenuated binding at acidic pH, enhanced transcytosis, and produced functional CNS readouts, such as neurotensin-induced hypothermia after peripheral administration, confirming pharmacodynamic data [103].

Fc region modification is another strategy. Reducing Fcγ effector function can mitigate reticulocyte depletion while preserving target effector engagement in the parenchyma when the therapeutic arm binds to its antigen [104]. Alternatively, FcRn-modulating variants can extend half-life or modestly increase brain exposure by altering intracellular trafficking [105]. FcRn binding also has limitations, as FcRn binding at neutral pH can shorten the half-life of an antibody and other Fc-containing proteins [106]. Human IgG has a half-life of approximately 19 to 21 days due to FcRn-mediated recycling, while conventional scFv or VHH antibodies typically have half-lives of only a few hours without specific half-life extension strategies [107]. However, by conjugating nanobodies with human albumin, the terminal half-life in primates can be extended to approximately 4.9 days, and the clearance rate is reduced.

Fragments and nanobodies reduce bulk and can minimise receptor cross-linking. However, they also reduce systemic half-life and, if not properly designed, can result in suboptimal distribution once inside the parenchyma [108]. The use of these small-volume modules requires consideration of how to extend half-life and carefully balance target and TfR affinity to prevent rapid washout after entering the target tissue [109].

### 4.3. Ligand-Based Strategies

#### 4.3.1. Transferrin Conjugation

Under physiological conditions, Tf enters brain endothelial cells by binding to TfR1 on the luminal membrane, triggering clathrin uptake and early endosome acidification [47]. When drugs are covalently linked to Tf via lysine-targeted carbodiimide conjugation, thiol-maleimide bridging to engineered cysteines, or bioorthogonal click reactions, the conjugates follow the same pathway [110]. Thus, Tf-bound complexes first bind to luminal TfR1 and then enter brain endothelial cells via endocytosis. The conjugates are then released within mature endosomes for abluminal export. Among lipid-based drug delivery approaches, Tf is commonly displayed on liposomes or polymeric particles via a DSPE-PEG linker, which keeps the ligand away from the bilayer and improves colloidal stability [111]. These carriers attach to the brain endothelium, transport through endosomes, and fuse outside the lumen, releasing their cargo into the perivascular space. Transferrin can also be linked to peptide particles, such as transferrin-Pep63 liposome, where Tf directs BBB trafficking while Aβ-binding peptides shape plaque-proximal pharmacology for clearance of oligomers and fibrils [112]. This is not only used in drugs for the treatment of AD, but also in Tf-modified liposomes for the delivery of chemotherapy drugs or genes to brain models for the treatment of glioma, as well as Tf-coupled vesicles to improve parenchymal exposure in the glioma and other cancer environments [113]. 

Given that Tf is an endogenous protein with broad tolerance, and the conjugate is compatible with scalable chemistries (EDC/NHS, maleimide, click), this facilitates binding to various carriers and reduces immunogenic responses. However, Tf needs to compete with abundant plasma transferrin, and random lysine conjugation interferes with iron binding or receptor recognition, which can lead to reduced transport efficiency [114]. At the same time, the formation of a protein corona on the nanoparticle may affect the direction of transport. To address these issues, some studies have adjusted the surface density of Tf and placed it on a flexible PEG tether to achieve the affinity required to enter endothelial cells without fixing the complex on the luminal surface [115,116].

Against other TfR1-targeted formats, transferrin conjugates generally achieve intermediate brain uptake with strong dependence on avidity and corona composition, and their systemic residence is governed by the carrier, which for many nanoparticles is in the hours range in rodents rather than the multi-week persistence typical of immunoglobulins that benefit from FcRn recycling in humans [117,118]. Off-target liabilities are dominated by competition with endogenous transferrin and reticuloendothelial clearance of decorated particles, whereas hematologic binding risks that concern some antibody shuttles are less central for these conjugates [118,119].

#### 4.3.2. Peptide Mimetics

Short TfR1-binding peptides can reduce spatial bulk while maintaining the same delivery route. On the luminal membrane, peptide ligands such as B6 bind to TfR1 and initiate clathrin uptake. Once inside the early endosome, the intermediate affinity and compact size increase the probability of dissociation, which favours entrapment of the complex into tubules and abluminal export. B6-modified PEG-PLA nanoparticles improve delivery of neuroprotective peptides in an AD-targeted design, where the B6 interface replaces full-length transferrin to drive endothelial uptake [120]. Based on this template, curcumin-loaded PLGA-PEG nanoparticles conjugated to B6 reached the brains of APP/PS1 mice, reduced Aβ deposition and tau phosphorylation, and improved cognition, suggesting that adjusting ligand affinity to the intermediate range and maintaining particle compactness shifts the complex from late endosomes toward transcytotic trafficking [121]. Other AD-relevant variants include selenium nanoparticles modified with B6 and sialic acid, which combine TfR1-mediated entry with anti-amyloid and antioxidant agents [122]. These case studies demonstrate the appeal of synthetic ligands in AD, as they avoid direct competition with endogenous Tf by binding to non-overlapping epitopes and can be manufactured in bulk with consistent valency. Their primary challenges are proteolysis, low per-site affinity, and residence time after entering the parenchyma [123].

Quantitatively, peptide systems often display greater inter-study variability in brain uptake than IgG-based shuttles because copy number, spacer length and carrier chemistry modulate effective avidity at the endothelial surface [124]. Their circulating half-life is usually short in the absence of additional engineering, commonly on the order of hours for peptide-decorated polymeric nanoparticles in rodents. Representative studies provide functional readouts in disease models, including curcumin-loaded PLGA–PEG nanoparticles conjugated to B6 that improved Morris water maze performance in APP/PS1 mice, as well as B6–sialic acid selenium nanoparticles that enhanced brain delivery and interfered with amyloid aggregation [120,125].

### 4.4. Positioning TfR1 Relative to Alternative BBB Receptors

Beyond modality-specific design, it is useful to position TfR1 against other receptor-mediated routes to clarify when it is preferable or complementary in AD. Multiple datasets indicate that endothelial TfR1 expression and TfR1-dependent internalisation remain largely preserved across disease windows in which other endothelial systems begin to fail. By contrast, insulin signalling in the cerebrovasculature is frequently altered in AD, with evidence of insulin resistance and selective receptor isoform loss that may reduce the stability of this route and introduces the risk of peripheral metabolic effects if agonism is engaged inadvertently [69]. LRP1 is a pivotal efflux receptor for amyloid beta and is widely expressed across the neurovascular unit, but endothelial LRP1 tends to decline with age and AD and competes with abundant endogenous ligands [65]. In addition, some LRP1-targeted cargoes are routed to lysosomes rather than recycled, which can limit throughput for large biologics [126].

These biological differences translate into practical consequences for engineering and safety. Insulin receptor shuttles must minimise agonism to avoid hypoglycaemia and desensitisation, often requiring format constraints that reduce apparent transcytosis capacity [127]. LRP1 shuttles can be effective in specific contexts but may face disease-stage variability in receptor density and higher rates of degradative routing, making dose scaling less predictable [128]. TfR1 shuttles, by contrast, exploit a receptor itinerary that is mapped in detail from early endosome to recycling, allowing designs that dissociate in acidic compartments, reduce receptor occupancy time and avoid competition with endogenous transferrin. The ability of a TfR1-shuttled anti-amyloid antibody to produce rapid and deep amyloid-PET clearance at low systemic dose in early symptomatic patients is consistent with a functional port of entry in vivo [56]. Taken together, TfR1 can serve as a primary entry route in AD, while the insulin receptor and LRP1 remain complementary options that can be layered to address regional heterogeneity, cargo biology or stage-specific needs.

### 4.5. Challenges in Translating TfR1-Targeted Therapies

Despite promising preclinical data, TfR1-mediated delivery strategies still have some limitations that need to be addressed. This section outlines several practical challenges and suggests directions for improvement. In vivo studies indicate that, although TfR1-mediated transport is a strong strategy for enhancing brain delivery of affinity binders, moderate-affinity binders often outperform very high-affinity binders for delivery to brain parenchyma [100]. This is because exogenous delivery systems must compete with endogenous transferrin for a limited number of TfR1 sites on the luminal membrane of brain endothelial cells; excessively high affinity can increase endothelial binding and saturate the transport pathway, yielding the classic bell-shaped exposure–affinity relationship. Microdialysis and physiologically based pharmacokinetic (PBPK) analyses corroborate that optimise affinity, valency, and pH-dependent unbinding broadens the therapeutic window [129]. Targeting non–transferrin-competing epitopes (e.g., the TXB2 VNAR epitope) can further improve distribution within brain parenchyma [130]. Early clinical trial (NCT04639050) demonstrate the feasibility of optimising the transport route, limiting receptor saturation, and reducing competition with transferrin [92]. Related receptor pathways remain under active clinical exploration, though with more heterogeneous evidence. Insulin-pathway trials in AD have chiefly used intranasal insulin rather than an insulin-receptor antibody shuttle. The pivotal SNIFF randomised trial (NCT01767909) did not meet its primary cognitive endpoint in the intention-to-treat population, even as pharmacologic delivery to brain was demonstrated and exploratory signals continue to motivate device and dosing refinements [131]. Furthermore, high affinity may lead to the antibody being retained within endothelial cells due to slow release kinetics, or being rapidly targeted for lysosomal degradation, thus affecting its ability to cross into brain tissue and bind to its target, ultimately limiting the effective amount of drug that reaches the brain [132]. The details of this transport mechanism are still unclear, which poses a significant challenge in research. The brain penetration rate of some monoclonal antibodies is typically less than 1.5% of the systemically administered dose [133]. 

TfR1 biology at the BBB is shaped by substantial regional and interindividual heterogeneity, which likely contributes to variability in drug distribution and pharmacodynamics. Although adult TfR expression is generally maintained with ageing, the impact of AD on BBB transport remains uncertain [134,135]. Data from in vivo models (e.g., 5xFAD) illustrated no marked reduction in endothelial TfR1 abundance in AD relative to age-matched controls [53]. However, preserved receptor levels do not guarantee intact function. Some disease-associated changes, such as chronic neuroinflammation, altered endothelial endocytosis and sorting, pH and redox shifts, or perivascular microenvironmental remodelling, which may impair receptor-mediated transcytosis independently of receptor expression. Consequently, affinity guided delivery platforms that rely on TfR1 may experience region specific efficiency and patient specific exposure, despite comparable nominal target expression [136]. This persistent uncertainty highlights a broader gap that the field lacks comprehensive and quantitative biomarkers that report BBB receptor expression and, critically, transport competence across the AD continuum. Addressing this gap will require harmonised human datasets spanning brain regions and disease stages, coupled with functional assays that distinguish receptor presence from trafficking capacity, so that TfR1-targeted therapeutics can be rationally dosed and better stratified for clinical response.

Finally, crossing the BBB via TfR generally requires systemic dosing, which introduces risks of off-target binding and peripheral toxicities. In clinical studies, high doses of TfR-targeted bispecifics, such as trontinemab, have been associated with infusion-related reactions (IRRs) and other systemic adverse events, largely early and mitigable with premedication [137]. Because reticulocytes abundantly express TfR1, interference with physiological iron uptake raises concerns about long-term safety [138]. Engineering strategies that lowering TfR-binding affinity and silencing Fc effector function can mitigate these risks while preserving brain delivery. Careful optimisation of affinity and epitope is therefore needed to minimise interaction with non-BBB TfR1.

### 4.6. Hybrid or Combination RMT Strategies

Building on these translational challenges, several hybrid approaches have been proposed to widen the therapeutic window and smooth regional heterogeneity in exposure. Rather than relying on a single receptor interface, future systems increasingly explore combinations that align with AD biology. Dual-receptor formats seek to couple the fast endothelial entry typical of TfR1 with complementary trafficking or distribution features of alternative routes, while dual-ligand nanoparticles and parallel, stage-tailored regimens aim to mitigate perfusion limits and regional receptor variability without inflating dose.

Based on these challenges, attention is now turning towards approaches that do not rely solely on a single receptor, but instead combine complementary transport systems to improve distribution. One avenue is the development of dual-receptor shuttles, in which TfR1 engagement is paired with a second pathway such as CD98hc or LRP1 [139]. Preclinical studies suggest that this strategy can couple the rapid endothelial entry characteristic of TfR1 with alternative trafficking dynamics that favour more sustained parenchymal exposure [140]. Yet these constructs require careful calibration of affinity and valency, as simultaneous receptor engagement risks competition or redirection to lysosomal degradation, which may offset the intended benefit [141].

A broader conceptual extension is to treat TfR1, the insulin receptor and LRP1 as complementary rather than competing options, each contributing to different contexts [8]. In early symptomatic AD, endothelial TfR1 remains the most stable and well-mapped interface, offering a reliable port of entry at stages when other BBB components are already impaired [82]. LRP1-based strategies may still provide value in regions where expression is preserved or for ligands that naturally exploit its itinerary, while insulin receptor–directed formats can be considered if metabolic safety is addressed [61]. This layered approach does not remove the engineering hurdles of receptor-mediated transcytosis, but it frames a realistic path that anchors drug entry in the preserved TfR1 pathway, deploying alternative receptors where they add unique value, and refining the molecular design to harmonise receptor engagement, intracellular trafficking and systemic tolerability.

Pairing TfR1 shuttling with nanoparticle formulations leverages complementary mechanisms at the endothelial and parenchymal levels. A TfR1-binding motif anchors the carrier to the luminal surface and triggers clathrin-mediated entry, while nanoscale engineering controls loading capacity, release kinetics, and dispersion within brain tissue. Studies with immunoliposomes targeting TfR1 show that endothelial accumulation can still translate into higher parenchymal drug exposure when payloads are released from endosomal compartments, which underscores the value of pairing shuttling with pH dependent release and endosomal escape modules [11]. Formation of a protein corona in biological fluids can mask transferrin ligands and blunt TfR1 engagement; therefore, surface chemistry and spacer design require experimental calibration to preserve ligand accessibility and to limit off-target clearance [118]. Disease-relevant examples include T7 peptide or transferrin-modified liposomes that raise brain exposure and improve functional outcomes in ischemic stroke and AD models, which provides in vivo evidence that receptor engagement and nanomaterial design can be aligned to produce additive effects across the barrier and within the parenchyma [142].

Bispecific architectures that couple a TfR1 shuttle arm with a therapeutic arm exemplify a parallel strategy in which barrier transport and target engagement are engineered within the same molecule yet can also be co-deployed with other vectors. Translation to higher species has been demonstrated with TfR1 with BACE1 constructs that cross the barrier and lower amyloid in nonhuman primates, while transport-vehicle platforms that graft TfR1 binders onto Fc or enzyme scaffolds improve brain exposure and enable pharmacodynamic effects such as broad distribution and substrate reduction for lysosomal enzymes [143,144]. Safety and developability considerations are integral to combination use, because peripheral TfR1 expression can lead to reticulocyte binding and dose-dependent liabilities, mitigation relies on lowering TfR1 affinity, enforcing monovalent engagement, attenuating effector functions, and managing receptor saturation during dosing, without compromising the therapeutic arm once the molecule reaches brain tissue [145]. These parameters are compatible with the co-administration of nanocarriers or small molecules that address complementary biology, thereby decoupling barrier transit from downstream pharmacology while maintaining acceptable systemic tolerability.

Peptide mimetics of TfR1 ligands provide a compact and programmable alternative that integrates readily with nanoparticle and antibody strategies. Short motifs such as B6 can be arrayed at controlled copy numbers on carrier surfaces, which facilitates diffusion through dense extracellular matrices and avoids Fc related interactions, while retaining productive receptor engagement and favourable endosomal release. In AD models, transferrin-Pep63 liposomes illustrate a layered mechanism in which TfR1 shuttling increases brain access and a second peptide domain captures oligomeric amyloid for enhanced clearance and rescue of synaptic plasticity [112,142]. Endosomal escape can be reinforced by co-displaying pH-sensitive fusogenic peptides, including GALA-derived sequences, which become membrane active in acidified compartments and thereby support the release of nucleic acids and other labile payloads into the cytosol or along recycling routes [146]. The principal limitations of peptide-based combinations arise from the same variables that govern nanoparticle performance, namely the nonmonotonic interplay of affinity and ligand density, the susceptibility of small ligands to corona masking, and peptide stability in plasma. These factors can be addressed through cyclization, D-amino acid substitution, and spacer optimisation while preserving the TfR1 engagement profile that favours transcytosis.

## 5. Conclusions

This review positions TfR1 as a compelling and resilient entry point for CNS therapeutics in AD. On the endothelial surface, TfR1 follows a well-characterised itinerary from binding to early endosome to recycling, which translates into predictable design rules for shuttles and ligand-based particles and helps maintain performance across ages and disease stages in which other barrier components are unstable. Evidence that receptor expression and receptor-dependent internalisation are preserved in the human cortex and isolated brain microvessels, and remain stable in age-related in vivo models, supports practical deployment for receptor-mediated delivery.

Looking forward, TfR1 plays a significant role as the anchor pathway in a multimodal therapeutic strategy. In this framework, receptor-enabled entry is combined with BBB repair therapies that stabilise the neurovascular unit, restore endothelial TJ integrity, support pericyte function and improve astrocytic polarity. Parallel use of neuroprotective drugs that reduce oxidative stress, normalise excitatory–inhibitory balance or modulate maladaptive microglial responses can address downstream injury cascades once brain access is secured. TfR1 shuttles can also be paired with disease-modifying interventions, including anti-amyloid and anti-tau biologics, nucleic acid therapies and enzyme or protein replacement, to couple barrier transit with parenchymal target engagement. Where regional heterogeneity or stage-specific biology limits distribution, complementary receptors such as the insulin receptor or low-density LPR1 can be layered in a context-dependent manner to widen coverage without abandoning the primary entry route.

From the application perspective, the same engineering principles that underpin TfR1 transport remain central in a multimodal setting. Monovalent binding with moderate apparent affinity and acidic endosomal dissociation reduces receptor occupancy time and favours recycling, enabling lower systemic dose and more flexible scheduling. Integration with quantitative biomarkers such as amyloid positron emission tomography, fluid markers of neurodegeneration and imaging of barrier integrity can guide adaptive dosing and stage-appropriate combinations. Although limitations persist, including potential competition in complex biological fluids and the need for rigorous control of ligand density and valency, ongoing advances are defining tractable solutions. Overall, TfR1-directed delivery represents a robust pathway across the BBB in AD and provides a practical scaffold on which barrier repair, neuroprotection and disease modification can be assembled into a coherent multimodal approach.

## Figures and Tables

**Figure 1 ijms-26-09793-f001:**
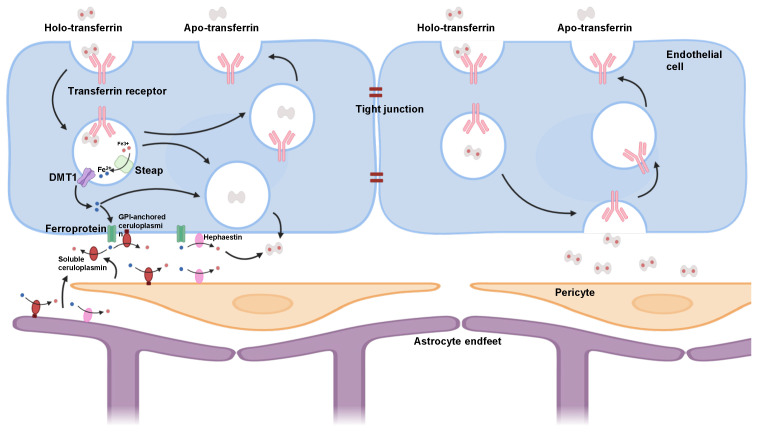
Schematic of transferrin receptor (TfR)-mediated transcytosis and iron handling at the blood–brain barrier (BBB). Brain microvascular endothelial cells form a selective, tight-junction sealed barrier between the blood (**top**) and abluminal side (**bottom**), closely ensheathed by pericytes. Holo-transferrin (Fe^3+^-Tf) selectively binds luminal TfR1. The complex undergoes clathrin-mediated endocytosis, endosomal acidification promotes iron release. This image was created with BioRender (https://www.biorender.com/ accessed on 5 October 2025).

**Figure 2 ijms-26-09793-f002:**
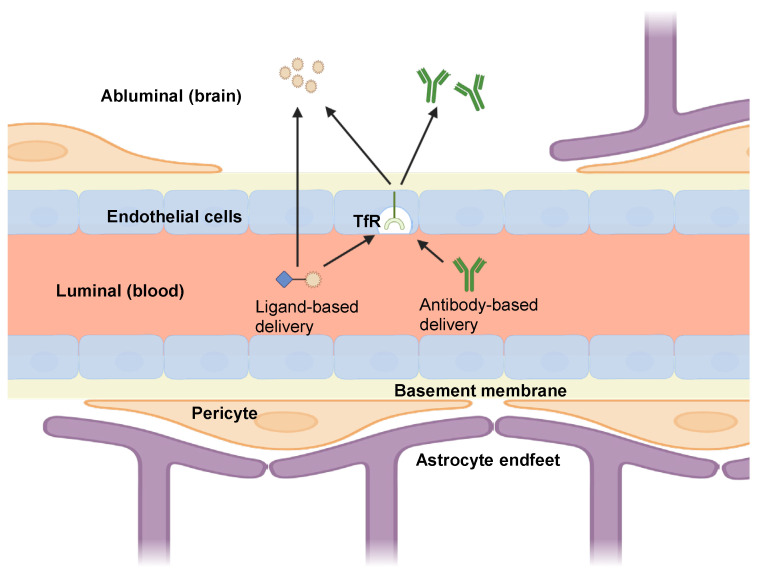
The schematic shows ligand- and antibody-based delivery binding transferrin-receptor (TfR) on endothelial cells, with complexes crossing the endothelial layer. Some ligand-mediated particles may traverse without engaging TfR. Effective TfR1 shuttles use monovalent, moderate-affinity binding, enable pH-dependent dissociation in early endosomes, target recycling-biased rather than degradation-prone epitopes, and control ligand density to avoid multivalent clustering and lysosomal capture. This image was created with BioRender (https://www.biorender.com/ accessed on 5 October 2025).

## Data Availability

No new data were created or analyzed in this study.

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
