# Peer review of "Targeting Transferrin Receptor 1 for Enhancing Drug Delivery Through the Blood–Brain Barrier for Alzheimer’s Disease"

_ijms, 2025, doi:10.3390/ijms26199793_

Round 1

Reviewer 1 Report

Comments and Suggestions for Authors

Manuscript Title: Targeting transferrin receptor 1 for enhancing drug delivery through the blood–brain barrier for Alzheimer’s disease. The manuscript addresses a highly relevant topic and provides a well-structured overview of transferrin receptor 1 (TfR1)-mediated strategies for overcoming the BBB in Alzheimer’s disease. This is an important contribution to the field of neurotherapeutics. However, there are several areas where the manuscript could be strengthened.

Major Comments:

1. Author should clearly emphasize the translational gap: while TfR1 targeting is promising, challenges such as receptor saturation, competition with endogenous transferrin, and intracellular trafficking inefficiencies limit clinical application.

2. Please elaborate more on the early onset/stages of AD before overt plaque/tangle pathology with recent report. For example

         https://pmc.ncbi.nlm.nih.gov/articles/PMC9107516/

         https://pubmed.ncbi.nlm.nih.gov/39046584/

3. Highlight how TfR1 stability during these early phases may make it a valuable therapeutic entry point.

4. Author should describe more explicitly the components of the BBB (endothelial cells, pericytes, astrocytic end-feet, basement membrane, tight junction proteins) in AD. For example

         https://pubmed.ncbi.nlm.nih.gov/37738628/

5. I would recommend elaborating on how oxidative stress, APOE4 status, and vascular dysfunction contribute to BBB changes that intersect with TfR1-mediated drug delivery.

6. Please elaborate on combination strategies where TfR1 shuttling can be used in parallel with other interventions (e.g., nanoparticles, bispecific antibodies, peptide mimetics).

7. Author should add more comparison with alternative receptor-mediated transport systems (e.g., insulin receptor, LRP1) and clarify why TfR1 might be superior or complementary.

8. I would recommend redrawing all the figures with higher clarity, consistent style, and explanatory legends.

9. A graphical abstract summarizing the TfR1-mediated delivery process in health vs AD would improve accessibility.

10. Please elaborate on ongoing or completed clinical trials targeting TfR1 or related pathways in AD, to highlight the translational readiness.

11. Author should explicitly acknowledge limitations of TfR1 targeting, including peripheral TfR1 binding, risk of off-target effects, and variability in iron metabolism in AD patients.

12. I would recommend expanding the conclusion to emphasize how TfR1-based strategies could be integrated with BBB repair therapies, neuroprotective drugs, or disease-modifying interventions to create a multimodal therapeutic approach.

Reviewer 2 Report

Comments and Suggestions for Authors

Dear Editors,

This review addresses an important and timely question: whether transferrin receptor 1 (TfR1) can be exploited to improve drug delivery across the blood–brain barrier (BBB) in Alzheimer’s disease (AD). The manuscript is clearly written, logically structured, and covers the biology of the BBB, TfR1’s role in iron transport, and a wide range of delivery strategies including antibodies, fragments, ligand conjugates, and nanoparticles. The illustrations (e.g., Figure 1 on page 5 showing TfR1-mediated uptake and transcytosis) are clear and help the reader.

The main strength is the comprehensive summary of receptor-mediated transport principles and their application to drug delivery. It is particularly helpful that the authors note that TfR1 expression appears preserved in AD models, distinguishing it from other BBB targets.

However, several issues reduce the overall impact and clarity. The review currently leans heavily on descriptive detail and summarises many individual studies, but the critical synthesis and comparison across strategies is limited. Important challenges are mentioned (competition with transferrin, receptor saturation, lysosomal trafficking), but there is little attempt to prioritise which are most limiting, or to outline design principles that the field agrees upon. In addition, while the focus is Alzheimer’s disease, much of the discussion draws on broader BBB literature without critically assessing whether AD-specific vascular pathology (heterogeneous BBB leakage, altered iron handling) alters the suitability of TfR1 targeting.

Major:

-The manuscript describes multiple approaches (bispecific antibodies, scFv fragments, nanobodies, ligand conjugates, nanoparticles), but the discussion is largely catalogue-like. Readers would benefit from a clearer evaluation of relative strengths and weaknesses, supported by direct comparison of key parameters such as brain uptake levels, half-life, and off-target effects. For example, antibody-based shuttles and peptide mimetics are both covered, but without any assessment of which currently shows the most translational promise.

-While it is emphasised that TfR1 expression is preserved in AD, other aspects of AD vascular pathology (localised BBB breakdown, iron accumulation, altered TfR1/Tf levels in cortex) are mentioned but not integrated into the conclusion. This leaves the reader unsure whether AD is indeed a particularly favourable context for TfR1 targeting, or whether these disease changes could confound delivery. A stronger conclusion would explicitly weigh these factors.

-The text notes that high-affinity bivalent binding diverts complexes to lysosomes, whereas moderate affinity, monovalent engagement, and pH-dependent dissociation favour transcytosis. However, this important principle is buried in the antibody section and not brought forward as a general design rule. A dedicated synthesis of mechanistic insights would improve the review’s utility.

-The review does not discuss recent advances in human in vitro BBB-on-a-chip or microvessel models that enable studying transport under flow, luminal/shear stress, imaging of transcytosis, and barrier integrity in more physiologically relevant settings. For example, human brain endothelial microvessel-on-a-chip or open microfluidic models are now able to reproduce key features of BBB transport and allow advanced optical imaging. Discussing these would strengthen the review by showing where TfR1 targeting has been or could be evaluated in these newer systems, and what they teach about transport kinetics, receptor saturation, and safety under more realistic conditions. Reference: https://pubmed.ncbi.nlm.nih.gov/33117784/

-The manuscript does not discuss whether any TfR1-based approaches have reached clinical trials in AD or related CNS diseases. Even if the answer is “none to date”, that absence itself is worth stating explicitly, as it frames the translational gap between strong preclinical rationale and clinical application.

-The review is narrowly focused on receptor-mediated transcytosis but does not discuss how TfR1-targeted strategies might intersect with glymphatic clearance pathways. Recent work has highlighted that impaired glymphatic function contributes to AD pathology and that tools now exist to measure glymphatic flow in both animal models and humans (see “Measuring glymphatic function: Assessing the toolkit”). Therapies designed to cross the BBB will ultimately be influenced by clearance through perivascular and glymphatic routes, which affect drug residence time and distribution. Integrating this perspective would add depth and situate TfR1-based delivery within the broader context of brain fluid dynamics.

Minor

-The text sometimes repeats content (e.g., the statement that TfR1 levels are preserved in AD is made in several places). This could be streamlined.

-A reference error (“Error! Reference source not found.” on page 8, antibody section) should be corrected.

-Some figure legends (e.g., Figure 2 on page 8) could be expanded to explain design principles, not just the schematic.

-The conclusion (page 10) states that TfR1 is “robust” for drug delivery, but does not acknowledge remaining uncertainties (competition with endogenous transferrin, iron dysregulation in AD). This should be addressed.

Reviewer 3 Report

Comments and Suggestions for Authors

The manuscript provides a well-structured and comprehensive review of Transferrin Receptor 1 (TfR1)–mediated drug delivery across the blood–brain barrier (BBB) in Alzheimer’s disease (AD). It is timely and relevant, given the urgent need for efficient CNS drug delivery strategies. The paper is strong in background coverage, use of recent references, and detailed mechanistic explanation. Figures are well chosen, and the writing is generally clear.

Abstract

  • Line 9–14: “offers an effective pathway to circumvent this barrier. Under normal physiology, TfR1 binds…” → Suggest rephrasing for smoother flow: “offers a potential pathway to circumvent this barrier. Physiologically, TfR1 binds…”
  • Line 19–21: Revise “competition with endogenous transferrin, potential receptor saturation, and suboptimal intracellular trafficking” → Suggest “competition with endogenous transferrin, receptor saturation, and inefficient intracellular trafficking.”

Introduction

  • Line 36–39: “One of the most significant reasons that hinders the development of AD therapy is the barriers of drug delivery.” → Revise for clarity: “One of the major challenges in AD therapy development is the difficulty of drug delivery to the brain.”

Section 3 (TfR1 biology and function)

  • Line 141–142: Contains “Error! Reference source not found.” → Please correct figure cross-referencing.
  • Line 147–149: Phrase “gatekeepers of the BBB” → better as “which act as gatekeepers of the BBB.”

Section 3.2 (TfR1 in AD)

  • Line 175–177: This idea (TfR1 preserved in AD) is repeated later; condense to avoid redundancy.
  • Line 203–205: “providing robust support…” → Suggest softening: “providing strong evidence supporting the potential of TfR1 as a vector target…”

Section 4 (Strategies and challenges)

  • Line 259–262: “while also addressing the associated challenges” → Could be rephrased: “and discusses the challenges associated with each approach.”
  • Line 301–302: Another “Error! Reference source not found.” → Fix figure reference.
  • Line 318–320: Sentences on monovalent vs. bivalent binding could be clarified; currently dense.
  • Line 385–400: Good explanation of peptide mimetics, but it ends abruptly. Suggest adding a sentence on their clinical translation status (any trials?).

Figures

  • Ensure all figure references are correct. Currently, Figure cross-links are broken (“Error! Reference source not found.”).
  • Captions are long; consider shortening them and moving mechanistic detail into the text.

Conclusion

  • Line 402–413: The conclusion is strong but repeats parts of earlier text. Suggest condensing and adding:
    • Future prospects (clinical trials, safety).
    • Limitations (competition with transferrin, off-target effects).
    • Open research questions (long-term modulation of iron homeostasis).

References

  • Check for formatting consistency: some references list DOIs as plain text, others inline.
  • Ensure all in-text citations match references.
Comments on the Quality of English Language

all comments attached for author 

Round 2

Reviewer 1 Report

Comments and Suggestions for Authors

I appreciate that authors have addressed all the comments meticulously. I whould recommend it for publication.

Reviewer 2 Report

Comments and Suggestions for Authors

Dear Editor,

The authors have successfully addressed the majority of my comments and concerns in order to improve the quality of the manuscript.

I believe that the new sections, improved ones, and updated references, have contributed to enhancing the clarity of the manuscript, which I can now endorse for publication.

All the best!

Reviewer 3 Report

Comments and Suggestions for Authors

Author responses are sufficient